# Experimental Investigation of Ground Radiation on Dielectric and Brightness Temperature of Soil Moisture and Soil Salinity

**DOI:** 10.3390/s20102806

**Published:** 2020-05-15

**Authors:** Weizhen Wang, Leilei Dong, Chunfeng Ma, Long Wei, Feinan Xu, Jiaojiao Feng

**Affiliations:** 1Key Laboratory of Remote Sensing of Gansu Province, Heihe Remote Sensing Experimental Research Station, Northwest Institute of Eco-Environment and Resources, Chinese Academy of Sciences, Lanzhou 730000, China; dongll@lzb.ac.cn (L.D.); machf@lzb.ac.cn (C.M.); weilong_gs@163.com (L.W.); xufeinan@lzb.ac.cn (F.X.); fengjiao@lzb.ac.cn (J.F.); 2Key Laboratory of Land Surface Process and Climate Change in Cold and Arid Regions, Chinese Academy of Sciences, Lanzhou 730000, China; 3University of Chinese Academy of Sciences, Beijing 100049, China

**Keywords:** dielectric model, brightness temperature, soil moisture, soil salinity, microwave radiometer

## Abstract

Soil moisture and salinity are crucial parameters of the Earth’s ecosystem; how to understand the radiation properties of them is of great significance for remote sensing monitoring. In this study, the application of mixed soil dielectric models (Dobson and generalized refractive mixing dielectric model (GRMDM)) and saline soil dielectric models (Dobson-S, HQR (Qingrong Hu), and WYR (Yueru Wu)) were analyzed to select the optimal models to simulate brightness temperature based on observational data. The brightness temperature of the soil moisture and multilevel salinity was simulated by using the Q-H (parameter of polarization mixing and parameter of characterizing height) model and Holmes parameterization scheme of soil effective temperature. The results show that both the Dobson model and the GRMDM model can well reproduce the real part and imaginary part of the dielectric constant of non-saline soil, and the GRMDM model was better. With the increase of the frequency, the simulation error of the dielectric constant of the saline soil by using the Dobson-S model, HQR model, and WYR model also increased, and the simulation result of the WYR model was better in the L band. The simulated result of the brightness temperature of soil moisture between the observation value and simulation value presented a high correlation both in the horizontal polarization and vertical polarization, with R greater than 0.967 and 0.948, and the root mean square error smaller than 3.998 K and 2.766 K, respectively. Meanwhile, the correlation coefficients of the brightness temperature of the saline soil in the horizontal polarization and vertical polarization were 0.935 and 0.971, and the root mean square errors were 5.808 K and 4.65 K, respectively. The brightness temperature decreased as the soil salinity increased, and the higher the salinity content was, the quicker the brightness temperature decreased. We expect that the experimental results can be used as a reference for algorithm developers to further enhance the accuracy of soil moisture and soil salinity retrievals.

## 1. Introduction

Soil moisture is an important element of hydrothermal delivery and energy transformation in the Earth’s ecosystem, and is not only the most active part of the regional water cycle, but also plays a critical role in hydrological, ecological, and biogeochemical processes [1,2,3]. Soil moisture is also a pivotal link of the cyclical process between the land surface and atmosphere and has been widely used as a key parameter in numerous environmental applications, including the estimation of crop yield [4], drought monitoring [5], modeling of land surface evaporation [6], and weather forecasting [7]. Therefore, soil moisture has been considered a vital observation parameter in the research of hydrology, meteorology, and agriculture, especially for environmental issues of arid and semi-arid regions, such as hydrometeorology, oasis agriculture, ecological environment, and sustainable development [8,9].

Soil saline-alkaline as a form of land degradation has become a dominant factor restricting regional economic growth, eco-environment protection, and agricultural sustainable development [10]. According to the incomplete statistics of The Food and Agriculture Organization of the United Nations (FAO), the soil saline-alkaline area has reached 954 million hm^2^, and this issue is more serious in arid and semi-arid regions [11]. Soil saline-alkaline are destructive for crop growth and production, they not only limit the ability of the root system to absorb moisture and nutrients, but also lead to crop yield declines and agricultural productivity fades when the salinity of soil reach high levels [12]. Finding out how to monitor soil saline-alkaline effectively, to obtain soil salinity content and detect spatial–temporal patterns of change, enabling effective measures to be taken to prevent soil saline-alkaline as soon as possible, is of great significance for improving crop yields and protecting the eco-environment.

There have been efforts made to monitor soil moisture and soil salinity all over the world. However, microwave remote sensing has the advantages of all-time and all-weather observation, strong penetrability, and being less affected by clouds, it has also been widely used for the monitoring of soil moisture [13,14,15] and soil salinity [16]. On the other hand, the main reason why microwave remote sensing can monitor soil moisture and soil salinity is that the brightness temperature and backscatter coefficient are closely related to soil dielectric property. The dielectric constant is mainly controlled by the contents of soil moisture and soil salinity [17,18]. Therefore, the dielectric model is the basis for retrieval of soil moisture and salinity using microwave remote sensing, and the empirical and semi-empirical dielectric models of non-saline-alkali soil such as the De Loor model [19], Topp model [20], Wang model [21], Hanllikainen model [22], Dobson model [23], and generalized refractive mixing dielectric model (GRMDM) [24], which are based on the theories of molar polarization, compact medium diffusion, and multiphase mixed medium have been widely used in soil dielectric research. However, those empirical and semi-empirical models ignored the effect of soil salinity. Some researchers have studied the response of complex permittivity to soil salinity through controlled experiments. Lasne [25] analyzed the influence of salinity on the dielectric constant by use of the soil moisture dielectric model. The result indicated that the real part of the dielectric constant decreased as the salinity content increased, while, the imaginary part increased. The analysis of the relationship between the complex dielectric constant and soil salinity content in the L band was conducted by Sreenivas [26] based on soil samples. The funding showed that the real part of the dielectric constant is hardly affected by salinity content, but the imaginary part increased as the salinity content increased. Shao [27] tested the dielectric constant of different soil moistures and soil salinities by the use of a network analyzer. The test found that the imaginary part of the dielectric constant changed obviously with the increase of soil moisture and soil salinity. Hu [28] and Wu [29] simulated the relationship between soil solution conductivity and salinity content, which was based on the Dobson model and Stogryn model [30]. As a consequence, a critical parameter of soil salinity content has been introduced into the expression of the imaginary part of the Dobson model by using conductivity. So far, there are no popular dielectric models of soil moisture and soil salinity in dielectric research, and the applicability of current dielectric models remains to be validated further.

As a potential monitoring method, the passive microwave remote sensing has been widely used in the detection of soil moisture at large scales [31,32] and has been used to form a series of general retrieval algorithms and forward models [33]. Monitoring of soil salinity using passive microwave remote sensing is still at the exploratory stage, but a few observation experiments of foundation and air-based microwave radiometer on radiation attributes of soil salinity have been conducted in recent years. Chaturvedi [34] conducted an observation test of brightness temperature in the field of river erosion. They claimed that the distinction of the effect between soil moisture and soil salinity was achieved by using the L band and C band together. A controlled experiment of soil moisture and soil salinity based vehicular microwave radiation system of Goddard Space Flight Center has been achieved by Jackson [35] in Beltsville Agricultural Research Center. The result shows that the land surface emissivity decreased as the soil salinity increased when soil moisture content was not changed. The soil moisture and soil salinity experiment of dual-polarization and multi-angle microwave radiation has been carried out by McColl [36] at the Nilpinna observation station in the south of Australia. They found that the error of soil moisture retrieval was more obvious if the influence of soil salinity was not considered for dielectric characteristics in salinization regions. Li [37] carried out a nondestructive observation of dual-polarization and multi-angle at bare saline-alkali land in the west of the Jilin province, China. The results showed that both the real part and the imaginary part of the dielectric constant in the L band and C band has a high correlation with soil moisture and soil salinity. In conclusion, the current research mainly focuses on the qualitative analysis of soil salinity to microwave radiation brightness temperature, and the promotion of the retrieval accuracy of soil moisture in saline-alkali land. Quantitative studies of soil salinity are still at the preliminary stage, the most observation experiments of soil salinity are carried out in natural saline-alkali land. Meanwhile, controlled experiments of the observation site of salinity have been seldom performed.

In this study, an observation experiment of microwave emission of soil moisture and multilevel salinity was conducted by using a ground-based microwave radiometer in Heihe remote sensing experimental research station of the Chinese Academy of Sciences. The objectives of this study are (1) to analyze the application of dielectric models of non-saline-alkali soil and saline soil; (2) to simulate the brightness temperature of soil moisture and multilevel soil salinity; (3) to research the influence of soil moisture and soil salinity on the brightness temperature; and (4) to provide a scientific foundation for effective simulation of the brightness temperature and quantitative retrieval of saline soil.

## 2. Materials 

A controlled observation experiment of soil moisture and soil salinity was conducted in Heihe remote sensing experimental research station of the Chinese Academy of Sciences from 11 to 19 May, 2018 (Figure 1). A multifrequency microwave radiometer of RPG-XCH-DP including L (1.4 GHz), Ku (18.7 GHz), and Ka (36.5 GHz) was exploited to measure the brightness temperature of H polarization and V polarization. The soil moisture, soil temperature, and dielectric constant of surface parameters were measured by Stevens Hydra Probe II, Campbell CS616, CSI109, and ML2X. The experiment was performed in the test field measuring about 20 m × 20 m based on the dual polarization microwave radiometer of RPG-XCH-DP (RPG-Radiometer Physics GmbH, Werner-von-Siemens-Str. 453340 Meckenheim, Germany). The main procedures for this experiment included fixing the microwave radiometer and the sensors that measured soil moisture and installing the soil temperature and dielectric constant, testing the physical parameters of soil samples, measuring the surface roughness, and measuring the brightness temperature of soil moisture and soil salinity.

The three observation transects were arranged with azimuth angles of 150°, 180°, and 210° according to the extent of the experiment and position of the microwave radiometer. The incident angle of the microwave radiometer was from 30° to 60° at a height of 6 m. The five moisture sensors of Stevens Hydra Probe II, four moisture sensors of CS616, five moisture sensors of ML2X, and four temperature sensors of CSI109 in this experiment were laid in the center of 8 pixels (Figure 2). When the azimuth was 180°, the moisture sensor of CS616 was laid in the pixel center where the incident angle was 30°. The moisture sensors of CS616, Stevens Hydra Probe II, and ML2X were arranged in the pixel center where the incident angle was 40°. The three temperature sensors of CSI109, which were used to measure the temperature of the soil depth at 5 cm, 10 cm, and 20 cm, were also placed here. The moisture sensors of CS616 and ML2X were placed in the pixel center where the incident angle was 50°. The moisture sensor of CS616 and the temperature sensor of CSI109 were placed in the pixel center where the incident angle was 60°. When the azimuth was 150° or 210°, the moisture sensors of Stevens Hydra Probe II and ML2X were placed in the pixel center where the incident angle was 40°. The moisture sensors of Stevens Hydra Probe II were placed in the pixel center where the incident angle was 50°. This observation experiment of soil moisture and soil salinity was divided into two stages: the first stage was the soil moisture observation, and the second stage was multilevel salinity observation. The NaCl aqueous salt solution was dissolved completely and was sprayed uniformly in the observation transects by means of fogging in the process of multilevel salinity observation; we then sprayed some water to ensure the NaCl aqueous salt solution could infiltrate to the depth of 5 cm. In the process of this experiment, we leveled the test field before the microwave radiometer installation to reduce the influence of ground surface roughness on radiation, and then measured the surface roughness by using the cross-section method. The measured result showed that the average root mean square height of the test field was 1.3 cm.

## 3. Methods

### 3.1. Soil Dielectric Model

The soil dielectric model is a critical parameter in the retrieval process of soil moisture and soil salinity. The theory of microwave remote sensing and the models of scattering and radiation in all tests are the function of the complex dielectric constant of the ground object [38,39]. The soil complex dielectric constant is a physical quantity, which brings electrode polarization under the influence of the external electric field. Usually, the ε of dielectric constant can present by the form of the complex number, which is expressed as follows:(1)$$\varepsilon =\varepsilon^{\prime }+\varepsilon^{\prime \prime }$$
where *ε*′ is the relative dielectric constant and *ε*″ represents the loss factor of the dielectric.

#### 3.1.1. Dielectric Model of Non-Saline-Alkali Soil

##### Dobson Model

The Dobson model [23] is a semi-empirical model of five different soil types based on dielectric constant system of waveguide and free-space propagation technique by taking observation data of soil samples in 1.4–18.7 GHz and four-component physical models into consideration. Peplinski [40] modified the Dobson model in 0.3–1.3 GHz to improve the application scope. As the most extensive dielectric model, the real part *ε*′*_soil_* and imaginary part *ε*″*_soil_* in the Dobson model are defined as follows:(2)$${\varepsilon^{\prime }}_{soil}={\left[1+\frac{\rho_{b}}{\rho_{s}}\left(\varepsilon_{s}{}^{\alpha }-1\right)+m_{v}{}^{\beta^{\prime }}{\varepsilon^{\prime }}_{fw}{}^{\alpha }-m_{v}\right]}^{1/\alpha }$$
(3)$${\varepsilon^{\prime \prime }}_{soil}={\left[m_{v}{}^{\beta^{\prime \prime }}{\varepsilon^{\prime \prime }}_{fw}{}^{\alpha }\right]}^{1/\alpha }$$
where *α* is the shape factor, *ρ_s_* is the density of soil particles, *ρ_b_* is the soil bulk density, *ε_s_* is the dielectric constant of soil solid phases, *m_v_* is the volume water content of soil, and *ε*′*_fw_* and *ε*″*_fw_* are the real part and imaginary part of the dielectric constant of soil free water, respectively, calculated by the Debye function [41]. *Β*′ and *β*″ are the adjustable parameters related to soil types, which are calculated by the sand content and clay content in soil. The detailed forms can be expressed as follows:(4)$$\beta^{\prime }=1.2748-0.159S-0.152C$$
(5)$$\beta^{\prime \prime }=1.33797-0.603S-0.166C$$

##### GRMDM Model

The generalized refractive mixing dielectric model (GRMDM) [24] was created by Mironov based on the refractive mixing dielectric model (RMDM) [42] and observation data in the range of 0.3–26.5 GHz. Mironov [43] obtained the parameter function between input parameters of model and clay content based on 15 different soil samples in order to extend the wide application of this model in 2009. According to the GRMDM model, the dielectric constant of soil with different water content can be drawn as follows:(6)$$\sqrt{\varepsilon_{s}}=\sqrt{\varepsilon_{d}}+(\sqrt{\varepsilon_{bw}}-1)V_{w}(V_{w}\leq V_{t})$$
(7)$$\sqrt{\varepsilon_{s}}=\sqrt{\varepsilon_{d}}+\left(\sqrt{\varepsilon_{bw}}-1\right)V_{t}+(\sqrt{\varepsilon_{fw}}-1)(V_{w}-V_{t})\;(V_{w}>V_{t})$$
where *ε_s_*, *ε_d_*, *ε_fw_*, and *ε_bw_* are the dielectric constant of soil, absolute dried soil, free water, and irreducible water, respectively; *V_ω_* is the volume water content of soil; and *V_t_* is the transitional water content of soil.

#### 3.1.2. Dielectric Model of Saline-Alkali Soil

##### Dobson-S Model

In the Dobson-S model, the real part *ε*′*_sw_* and the imaginary part *ε*″*_sw_* of the dielectric constant of the saline solution replace the real part *ε*′*_fw_* and the imaginary part *ε*″*_fw_* of the dielectric constant of free water. The detailed forms are defined as follows:(8)$${\varepsilon^{\prime }}_{soil}={\left[1+\frac{\rho_{b}}{\rho_{s}}\left(\varepsilon_{s}{}^{\alpha }-1\right)+m_{v}{}^{\beta \prime }\varepsilon^{\prime }{}_{sw}{}^{\alpha }-m_{v}\right]}^{1/\alpha }$$
(9)$${\varepsilon^{\prime \prime }}_{soil}={\left[m_{v}{}^{\beta^{\prime \prime }}\varepsilon^{\prime \prime }{}_{sw}{}^{\alpha }\right]}^{1/\alpha }$$

##### HQR Model

The HQR (Qingrong Hu) model [28] modifies the imaginary part of the Dobson model and fits the relationship between soil solution conductivity and salt content so that the critical parameter of soil salt content can be introduced into the imaginary part of Dobson model by the use of conductivity. The conductivity of the soil salt solution can be presented as follows:(10)$$\sigma =0.14A\frac{(\rho_{s}-\rho_{b})\rho_{b}}{\rho_{s}}\frac{S}{m_{v}{}^{2}}\chi$$
where *χ* is the temperature compensation coefficient; the value of A depends on salt ion types of solution; *ρ_s_* is the density of soil particles; *ρ_b_* is soil bulk density; *S* is soil salt content; and *m_v_* is the volume water content of soil. The real part *ε*′*_sw_* and imaginary part *ε*″*_sw_* of the dielectric constant in HQR model are defined as follows:(11)$${\varepsilon^{\prime }}_{sw}=\varepsilon_{sw\infty }+\frac{\varepsilon_{sw0}-\varepsilon_{sw\infty }}{1+{\left(2\pi f\tau_{sw}\right)}^{2}}-k_{0}\varepsilon_{sw0}S_{m_{v}}\approx \varepsilon_{sw\infty }+\frac{\varepsilon_{sw0}-\varepsilon_{sw\infty }}{1+{\left(2\pi f\tau_{sw}\right)}^{2}}$$
(12)$${\varepsilon^{\prime \prime }}_{sw}=\frac{2\pi f\tau_{sw}(\varepsilon_{sw0}-\varepsilon_{sw\infty })}{1+{\left(2\pi f\tau_{sw}\right)}^{2}}-\left(2k_{0}\tau_{sw0}S_{m_{v}}\cdot f\right)+\frac{A\zeta \chi }{2\pi \varepsilon_{0}}\cdot \frac{\left(\rho_{s}-\rho_{b}\right)\rho_{b}}{\rho_{s}}\cdot \frac{S}{m_{v}{}^{2}}\cdot \frac{1}{f}$$

##### WYR Model

The WYR (Yueru Wu) model [28] expresses the relationship between soil solution conductivity and the salt content based HQR model as follows:(13)$$\sigma_{w}\approx c\cdot \rho_{b}\frac{{\left(S\right)}^{m_{v}{}^{\alpha }}}{m_{v}}$$
where *c* and *α* are fitted by observation data, *c* is 0.371, and *α* is 0.18. the real part *ε*′*_sw_* and imaginary part *ε*″*_sw_* of the dielectric constant in WYR model are defined as follows:(14)$${\varepsilon^{\prime }}_{sw}=\varepsilon_{sw\infty }+\frac{\varepsilon_{sw0}-\varepsilon_{sw\infty }}{1+{\left(2\pi f\tau_{sw}\right)}^{2}}$$
(15)$${\varepsilon^{\prime \prime }}_{sw}=\frac{2\pi f\tau_{sw}(\varepsilon_{sw0}-\varepsilon_{sw\infty })}{1+{\left(2\pi f\tau_{sw}\right)}^{2}}-0.371\cdot \frac{\rho_{b}}{2\pi f\varepsilon_{0}}\cdot \frac{{\left(S\right)}^{m_{v}{}^{0.18}}}{m_{v}}$$

### 3.2. Emissivity Model of Rough Surface

The common semi-empirical emissivity models of rough surface are the Hp model [44], Q-H (parameter of polarization mixing and parameter of characterizing height) model [45], and Qp model [46]. The widest used application of the semi-empirical emissivity model of exposed soil is that they can present the effective emissivity of the exposed surface. The form of the Q-H model is defined as follows:(16)$$r_{Gp}(\theta )=[(1-Q_{R})\times r{*}_{Gp}(\theta )+Q_{R}r{*}_{Gq}(\theta )]\times \exp(-H\cos^{2}(\theta ))$$
where *θ* is the incident angle; *r***_G_* and *r_G_* are emissivity of the smooth and rough surface, respectively; *p* and *q* are the horizon polarization factor and vertical polarization factor, respectively; *H* is the roughness parameter; and *Q_R_* is the mixed polarization factor.

### 3.3. Soil Effective Temperature

The soil effective temperature is the microwave radiation sum of the whole soil layer according to the microwave radiative transfer theory [47]. It is difficult to obtain observation parameters of multilayer soil to calculate the soil effective temperature based on the theoretical method. Choudhury [44], Wigneron [48], Holmes [49], and LV [50] have put forward successive parameterization schemes to obtain the soil effective temperature effectively. Ma [51] analyzed the sensitivity of soil parameters to brightness temperature, and found that the sensitive difference was mainly caused by different parameterization schemes. They also claimed that the Holmes parameterization scheme was more reasonable. The Holmes parameterization scheme is accepted to calculate soil effective temperature in this study. The forms are defined as follows:(17)$$T_{eff}=T_{Deep}+C\left(\varepsilon \right)(T_{Surf}-T_{Deep})$$
where *T_eff_* is the soil effective temperature, *T_surf_* is the soil surface temperature at a depth from 0 to 5 cm, *T_deep_* is the soil deep temperature at a depth from 50 to 100 cm, and *C*(*ε*) is a parameter related to the dielectric constant, which is drawn as follows:(18)$$C(\varepsilon )={\left((\varepsilon^{\prime \prime }/\varepsilon^{\prime })/\varepsilon_{0}\right)}^{b}$$
where *ε*_0_ = max(*ε*″/*ε*′) and *b* is 0.9.

### 3.4. Brightness Temperature Simulation

As this experiment was carried out in bare soil, the brightness temperature of microwave radiation in this experiment comes from the whole soil layer and the atmosphere. The influence of atmospheric radiation on the brightness temperature can be ignored because the installation height of the microwave radiometer is near to the land surface. The brightness temperature is defined as follows:(19)$$TB_{P}\left(\theta \right)=T_{eff}\cdot \left(1-R_{P}\left(\theta \right)\right)$$
where *TB_p_* is the brightness temperature, *θ* represents the incident angle, *p* is the polarization mode, *R_p_*(*θ*) is the rough surface emissivity, and *T_eff_* is the soil effective temperature.

With all above-mentioned methods, our proposed methodological framework can be conducted. The work flowchart is schematically shown in Figure 3.

## 4. Results

### 4.1. Dielectric Constant Comparison of Non-Saline-Alkali Soil

The real part and imaginary part of the dielectric constant of the non-saline-alkali soil in the L (1.4 GHz) band and Ku (18.7 GHz) band were simulated by using the Dobson model and GRMDM model in this research. The results show (Figure 4, Table 1) that in both the L (1.4 GHz) band and Ku (18.7 GHz) band, the simulated values of the real part and imaginary part of the dielectric constant increased as the soil moisture content increased. The range of the simulated values in the real part was higher than the imaginary part. The simulated values decreased in the real part and increased in the imaginary part as the frequency increased on the condition that the factors of temperature, soil bulk density, sand content, clay content, and volume water content remained the same. The simulated values of the dielectric constant of the real part based GRMDM model were higher than that of the Dobson model in the L (1.4 GHz) band, but the simulated values of the Dobson model in the imaginary part were higher than that of the GRMDM model if the soil volume water content was less than 0.4 cm^3^/cm^3^. When the soil volume water content was greater than 0.4 cm^3^/cm^3^, the simulated values of the Dobson model were lower. In the Ku (18.7 GHz) band, the simulated values of the GRMDM model, both in the real part and imaginary part, were higher than that of the Dobson model.

The application of the Dobson model and the GRMDM model in the L (1.4 GHz) band and the Ku (18.7 GHz) band was achieved by comparing the real part and the imaginary part of the dielectric constant of the simulation values and the observation values of non-saline-alkali soil. The results in Table 2 show that the Dobson model and GRMDM model can simulate the real part and imaginary part of the dielectric constant of non-saline-alkali soil well. The R values for both the Dobson model and GRMDM model were 0.999 in the real part of the L (1.4 GHz) band and the Ku (18.7 GHz) band. The RMSE values of the Dobson model in the L (1.4 GHz) band and the Ku (18.7 GHz) band were 0.510 and 1.462, respectively, and the RMSE values of the GRMDM model were 0.503 and 1.433, respectively. In the imaginary part, the R values of the Dobson model in the L (1.4 GHz) band and Ku (18.7 GHz) band were 0.871 and 0.910, respectively, and the RMSE values were 2.631 and 2.524, respectively. The R values of the GRMDM model in the L (1.4 GHz) band and Ku (18.7 GHz) band were 0.894 and 0.895, respectively, and the RMSE values were 2.430 and 1.856, respectively. The errors of the GRMDM model both in the L (1.4 GHz) band and the Ku (18.7 GHz) band were smaller than for the Dobson model. Therefore, the GRMDM model was used to simulate the non-saline-alkali soil brightness temperature of the L (1.4 GHz) band in the next step.

### 4.2. Dielectric Constant Comparison of the Saline-Alkali Soil

Simulations of the imaginary part of the saline-alkali soil dielectric constant in the L (1.4 GHz) band and Ku (18.7 GHz) band were achieved by the use of the Dobson-S model, HQR model, and WYR model in this research, respectively. The trend and simulated results of the imaginary part with the changes of soil moisture are drawn in Figure 5 and presented in Table 3 when the soil salinity contents were 5 ‰, 10 ‰, 15 ‰, and 20 ‰. The results show that the simulated values of the Dobson-S model and the WYR model in the L (1.4 GHz) band increased as the soil volume water content increased. On the contrary, the simulated values of the HQR model decreased, and the amplitude was stable. When the soil salinity contents were 5‰ and 20‰, the changes of the maximum and minimum simulated by the Dobson-S model in the imaginary part were 7.89 and 2.289, respectively, while for the HQR model were 29.393 and 20.677, respectively, and for the WYR model they were 34.821 and 4.413, respectively. In the Dobson-S model, the real part and imaginary part of the dielectric constant of the free water were replaced of the saline solution. The sensitivity of the salinity is lower for the dielectric constant in the Dobson-S model. The soil salinity content was introduced into the imaginary part by the use of conductivity in the HQR model and the WYR model to improve the sensitivity of the dielectric constant imaginary part. The simulated values of the Dobson-S model, HQR model, and WYR model in the Ku (18.7 GHz) band increased as the soil volume increased. The amplitude of the Ku (18.7 GHz) band was smaller than that of the L (1.4 GHz) band.

The simulated results of the imaginary part with soil salinity changes when the soil volume water contents were 0.1 cm^3^/cm^3^, 0.2 cm^3^/cm^3^, 0.3 cm^3^/cm^3^, and 0.4 cm^3^/cm^3^ are illustrated in Figure 6 and Table 4. The simulated values of the Dobson-S model and the WYR model in the L (1.4 GHz) band increased as the soil moisture and the soil salinity increased. When the soil moisture was constant, the simulation of the HQR model increased as the soil salinity increased. However, the simulated changes of the HQR model decreased gradually as soil moisture increased. When the soil volume water content was 0.1 cm^3^/cm^3^, the minimum values of the Dobson-S model, HQR model, and WYR model were 0.251, 2.046, and 1.104, respectively, and the maximum values were 3.893, 48.879, and 8.591, respectively. When the soil volume water content was 0.4 cm^3^/cm^3^, the minimum values of the Dobson-S model, HQR model, and WYR model were 1.498, 2.456, and 4.007, respectively, and the maximum values were 12.151, 36.815, and 46.641, respectively. The simulated values of the HQR model were more sensitive when the soil volume water contents were smaller. However, the simulated values of the WYR model were more sensitive to soil salinity changes when the soil volume water contents were greater. The simulated values of the Dobson-S model, HQR model, and WYR model increased as the soil salinity content increased in the Ku (18.7 GHz) band. The change range of the Ku (18.7 GHz) band was smaller than that of the L (1.4 GHz) band because the wavelength of the Ku (18.7 GHz) band is shorter and its penetrability is weaker.

The inaccuracies of the simulation values based on the Dobson-S model, HQR model, and WYR model in the L (1.4 GHz) band and Ku (18.7 GHz) band were examined by a comparison of the observation values. The results revealed in Table 5 show that there was a high correlation between the measured values and the simulated values, and the root mean square error (RMSE) increased gradually as the frequency increased in the real part of the dielectric constant, and the values of it in the L (1.4 GHz) band and Ku (18.7 GHz) band were 3.240 K, and 7.721 K, respectively. There was a great difference of RMSE in the imaginary part. The RMSE of the Dobson-S model, HQR model, and WYR model in the L (1.4 GHz) band was 8.702 K, 11.159 K, and 4.508 K, respectively, and the errors in the Ku (18.7 GHz) band were 10.448 K, 9.086 K, and 9.475 K, respectively. The error in the WYR model was smaller than in the Dobson-S model and HQR model. Therefore, the WYR model was used to simulate the saline-alkali soil brightness temperature of the L (1.4 GHz) band in the next step.

### 4.3. Brightness Temperature Simulation of the Non-Saline-Alkali Soil

In this research, the application of dielectric models of non-saline-alkali soil and saline soil were analyzed to select the optimal dielectric models; then, the optimal dielectric models were used to simulate the brightness temperature of the soil moisture and multilevel soil salinity. The brightness temperature received by the microwave radiometer was influenced by the whole soil layer microwave radiation and the atmospheric radiation. The interference of atmospheric radiation for the brightness temperature can be ignored because the installation height of the microwave radiometer is near the surface. The surface emissivity and soil effective temperature can be computed by using the Q-H model and the Holmes parameterization scheme, respectively. The GRMDM model was the dielectric model used in the process of the brightness temperature simulation, and the root mean square height was 1.3 cm. The simulated results of the brightness temperature of non-saline-alkali soil are show in Figure 7. The results indicate that there was a high correlation between measured values and simulated values, and the correlation coefficients of H polarization and V polarization were 0.967 and 0948, respectively and the RMSE were 3.998 K and 2.766 K, respectively.

### 4.4. Brightness Temperature Simulation of the Multilevel Saline-Alkali Soil

The simulated method of the brightness temperature of the saline-alkali-soil was similar to that for the non-saline-alkali soil. The WYR model was regarded as the dielectric model, and the Q-H model and Holmes parameterization scheme were accepted to calculate the rough surface emissivity and the soil effective temperature, respectively. Finally, the brightness temperatures of the multilevel saline-alkali soil were simulated and are shown in Figure 8. The results indicate that the simulated brightness temperature both of H polarization and V polarization decreased gradually as the salinity content increased, and the higher the salinity content was, the greater the decreasing amplitude of brightness temperature became. The brightness temperature trends of different salinity contents were similar both in H polarization and V polarization. The analysis of observed values and simulation values can be achieved by using statistical methods, and the results showed that there was a high correlation between measured values and simulated values, and the correlation coefficients of H polarization and V polarization were 0.935 and 0.971, respectively, while the RMSEs were 5.808 K and 4.65 K, respectively.

## 5. Discussion

As the results have shown, both the Dobson model and GRMDM model can well reflect the dielectric constant of the non-saline-alkali soil. It is necessary to investigate the error source of the dielectric constant simulation. The Dobson model, based on the waveguide and free-space transmission technique, was constructed on the basis of a large number of measurements of the soil dielectric constant made using five kinds of soil. The dielectric property of bound water was introduced into the GRMDM model based on the Debye equation to perfect the dielectric model in principle, and it was also regarded as a single component in the model. Therefore, the accuracy of the GRMDM model was higher than the Dobson model. However, both the Dobson model and the GRMDM model only took into account the dependence of the dielectric behavior on soil moisture ignoring the effect of soil salinity. In the course of the preparation of the soil samples, the mineral ions of soil can dissolve in the soil solution, and the low frequency bands of the microwave are highly sensitive to the variation of soil salinity. Thus, the simulated results of the Ku (18.7 GHz) band were better than that of the L (1.4 GHz) band in the imaginary part.

There are obvious differences in the dielectric models of the saline-alkali soil. In the Dobson-S model, the real part and imaginary part of the free water were replaced by those of the salt water, and the property of the dielectric constant is different between the salt water and the saline soil. Therefore, the simulated errors of the Dobson-S model were higher as the soil moisture increased. The HQR model modified the expression of the imaginary part by considering a salinity factor to improve the Dobson semiempirical soil dielectric mixing model, based on the measurement made of the dielectric properties. However, Hu noticed that this model only performs well in high soil moisture conditions (*m_v_* ≥ 0.3 cm^3^/cm^3^), there is a significant remarkable error at the low frequency bands when soil moisture is low. Therefore, the simulated result of the HQR model in the L (1.4 GHz) band decreased as soil moisture increased. The WYR model improved the accuracy of the dielectric model of the saline soil under different soil moisture conditions, but the parameter of saturation was ignored in those dielectric models of the saline soil, and further research is needed to improve the applicability.

Surface roughness is a critical parameter that influences the accuracy of the brightness temperature simulation. Surface roughness is considered to increase the soil emissivity due to the increase in the surface area of the emitting surface. The Q-H model, as an emissivity model, has been widely used in soil moisture and soil salinity algorithms. The “Q” and “H” are two factors in the Q-H model, which are used to describe the influence of surface roughness. The two roughness factors can be featured by the surface root mean square and horizontal roughness correlation length in theory. However, it is difficult to quantify these two factors using observation data at the large-scale, especially in some special areas. Furthermore, the incidence angle and frequency also have an impact on the roughness factor. Due to the lack of measurement data to quantify “Q” and “H” around the world, the two parameters have to be assumed as constant values based on some typical experiments. In other words, “Q” and “H” are usually determined empirically. Therefore, the simulation errors of brightness temperature might be introduced if the empirical values of the two parameters in the Q-H model are inconsistent with the actual surface.

Furthermore, this observation experiment was conducted in the midstream of the Heihe river basin, which is located in the northwest of China, and the climate type in this area is arid and semi-arid. In summer, both the solar radiation and evaporation will reach the maximum values for the whole year. In the process of the multilevel salinity observation, the NaCl aqueous salt solution dissolved completely and was sprayed uniformly in the observation transects by means of fogging. It was then sprayed with some water to ensure that NaCl aqueous salt solution could infiltrate to a depth of 5 cm. When the solar radiation was strong enough in the daylight, the soil salinity could be brought to the surface with the evaporation of the soil moisture and finally covered the surface of the soil. This results in a simulation error of the brightness temperature because the soil salinity is higher at the soil surface.

Soil salinity is one of the critical parameters that influence the physical properties of soil [52]. The main reason why some serious issues of land degradation, such as clay contents dispersion, void plugging, and soil degeneration, always occur in some specific regions is that the sodium contents and exchangeable sodium ions are at high levels in the soil in those areas [53]. Those environmental problems have a detrimental impact on the growth of crops [54]. Therefore, the NaCl aqueous salt solution was used in this multilevel soil salinity experiment. However, other soluble salt ions of the soil can affect the soil’s physical properties and agricultural production. As we only considered the influence of the NaCl aqueous salt solution and ignored the impact of other soluble salt ions, the simulated error between observed values and simulated values was caused. Furthermore, we sprayed the water and NaCl aqueous salt solution in the test field by manual means to retain the soil moisture and soil salinity as much as possible, but this could not achieve the uniformity of an ideal state. It is also an unavoidable drawback in this observation experiment of soil moisture and soil multilevel salinity.

## 6. Conclusions

In this paper, the microwave observed experiment of soil moisture and multilevel salinity was conducted by using a ground-based microwave radiometer at the Heihe Remote Sensing Station. The application of the dielectric models of the mixed soil and saline-alkali soil were analyzed respectively based on the observed data of the microwave radiometer. The brightness temperature of soil moisture and multilevel soil salinity were simulated by using the Q-H model and Holmes parameterization scheme. The main conclusions can be summarized as follows:Both the Dobson model and GRMDM model can simulate the real part and imaginary part of dielectric constant well in the mixed soil, but the simulated error of the GRMDM model was smaller. Therefore, the GRMDM model was introduced to simulate the brightness temperature of non-saline soil in the L (1.4 GHz) band.There are obvious differences in the dielectric models of the saline-alkali soil, and the disadvantage exists in each model. The simulated error of the Dobson-S model, HQR model, and WYR model increased as the frequency increased. The simulated result of the WYR model in the L (1.4 GHz) band was better than that of the Dobson-S model and HQR model. Therefore, the WYR model was used to simulate the brightness temperature of the multilevel saline soil.The simulated result of the brightness temperature of soil moisture presented a high correlation, and the correlation coefficients of H polarization and V polarization were 0.967 and 0948, respectively, and the RMSEs were 3.998 K and 2.766 K, respectively.The simulated result of the brightness temperature of the multilevel saline soil demonstrated that the brightness temperature decreased gradually as the soil salinity increased both in H polarization and V polarization. The higher the soil salinity was, the greater the decreasing amplitude of the brightness temperature became. The correlation coefficients of H polarization and V polarization were 0.935 and 0.971, respectively, and the RMSEs were 5.808 K and 4.65 K, respectively.

## Figures and Tables

**Figure 1 sensors-20-02806-f001:**
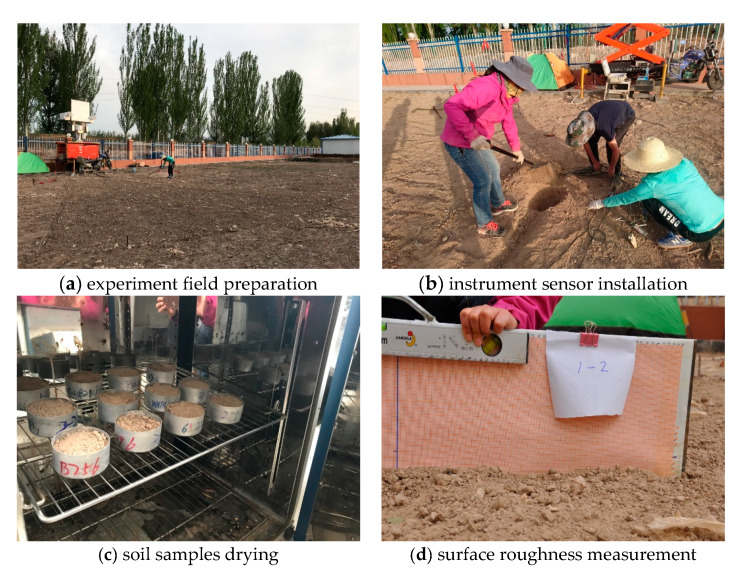
The observation experiment of soil moisture and soil salinity.

**Figure 2 sensors-20-02806-f002:**
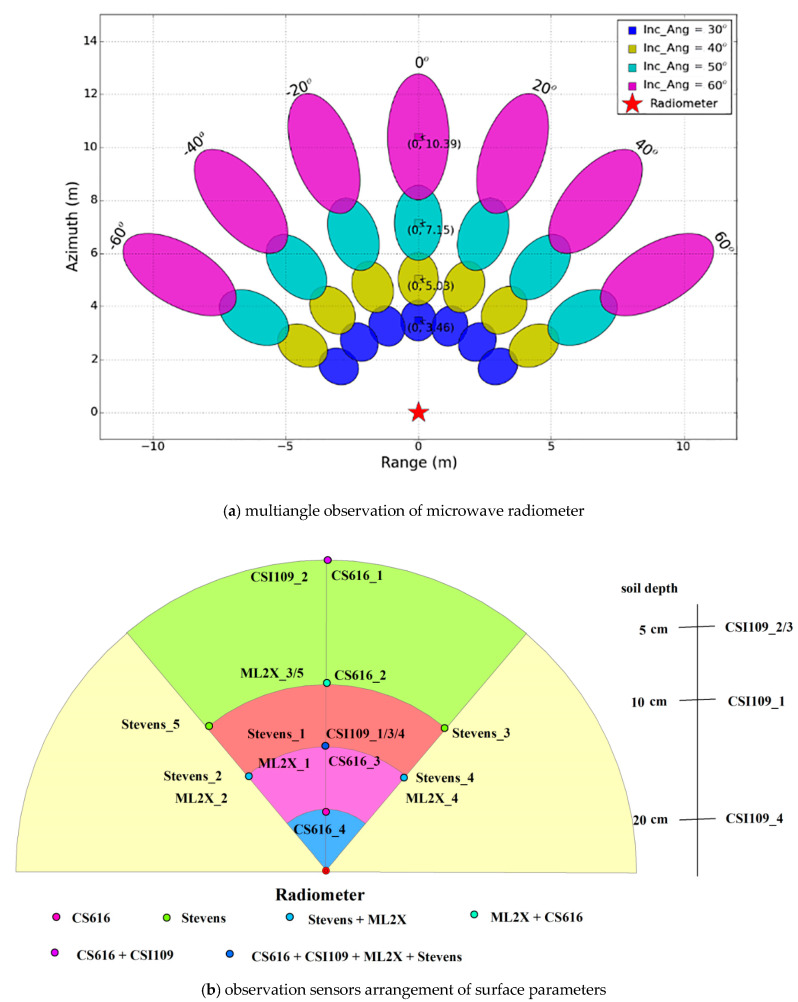
The experiment layout of soil moisture and soil salinity.

**Figure 3 sensors-20-02806-f003:**
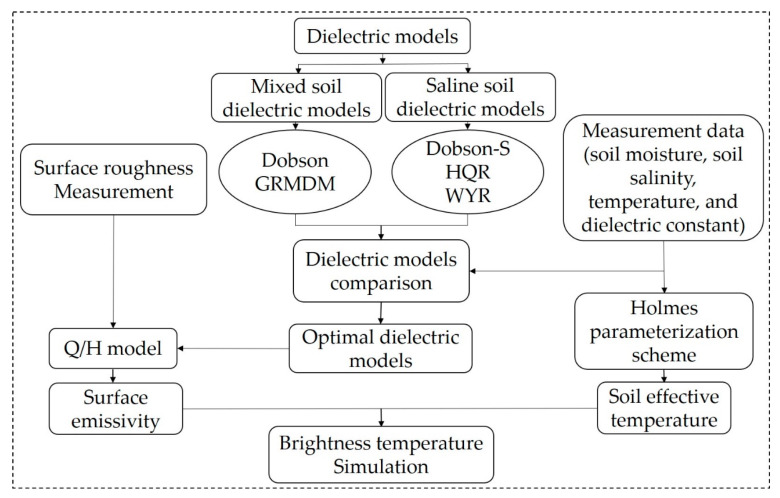
Flowchart of the proposed methodological framework for the brightness temperature simulation.

**Figure 4 sensors-20-02806-f004:**
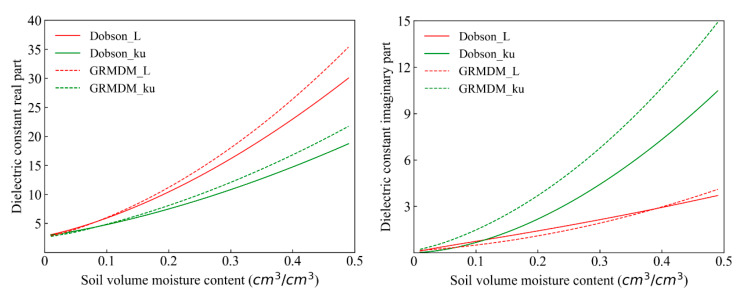
The dielectric constant of non-saline-alkali soil varies with soil moisture.

**Figure 5 sensors-20-02806-f005:**
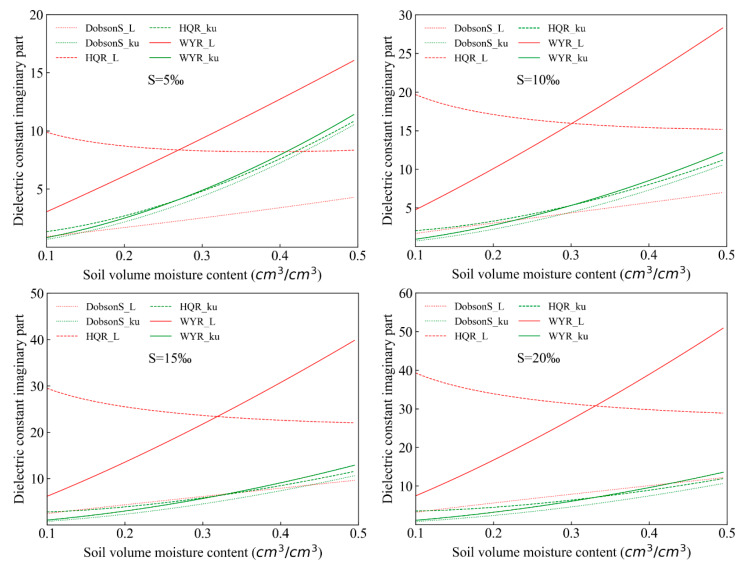
The imaginary part of the dielectric constant varies with soil volume moisture content in the saline-alkali soil.

**Figure 6 sensors-20-02806-f006:**
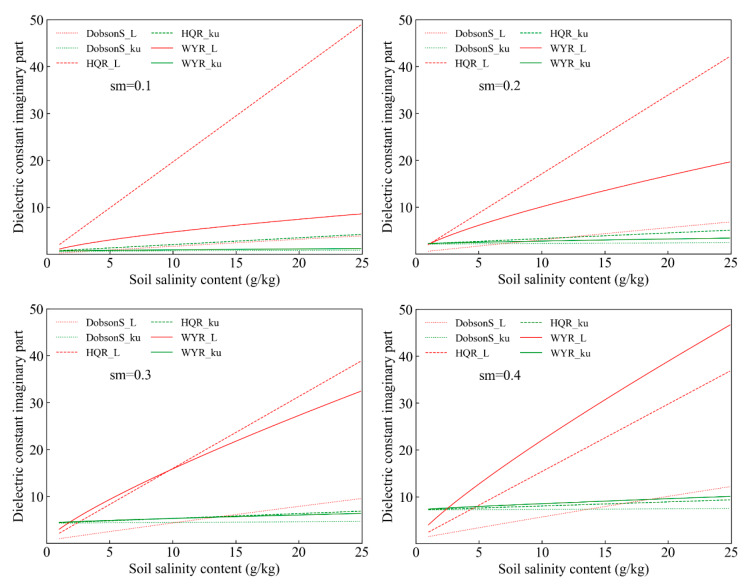
The imaginary part of the dielectric constant varies with the soil salinity content.

**Figure 7 sensors-20-02806-f007:**
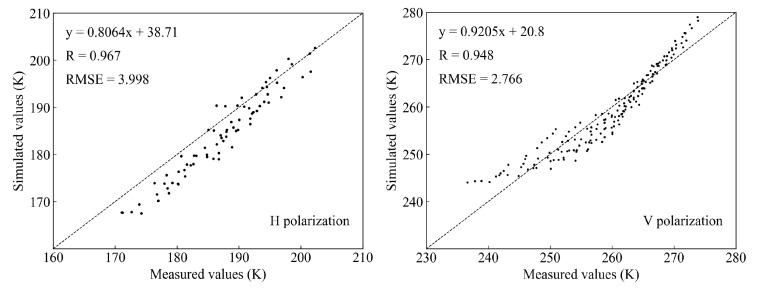
The simulation of brightness temperature of the non-saline-alkali soil at the L band.

**Figure 8 sensors-20-02806-f008:**
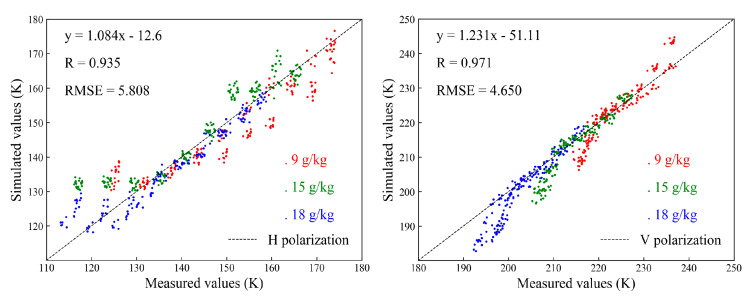
The simulation of brightness temperature of the saline-alkali soil at the L band.

**Table 1 sensors-20-02806-t001:** The simulation of dielectric constant of non-saline-alkali soil.

Soil Moisture (cm^3^/cm^3^)	The Real Part				The Imaginary Part			
	Dobson-L	Dobson-Ku	GRMDM-L	GRMDM-Ku	Dobson-L	Dobson-Ku	GRMDM-L	GRMDM-Ku
0.01	3.063	2.992	2.839	2.757	0.110	0.018	0.148	0.217
0.1	5.914	4.783	5.998	4.850	0.736	0.659	0.492	1.469
0.2	10.422	7.486	11.199	8.083	1.411	2.177	1.091	3.699
0.3	16.140	10.817	18.010	12.072	2.138	4.420	1.915	6.766
0.4	22.982	14.731	26.431	16.816	2.930	7.324	2.963	10.670
0.5	30.049	18.727	35.358	21.723	3.699	10.469	4.099	14.899

**Table 2 sensors-20-02806-t002:** The errors analysis of the dielectric constant of non-saline-alkali soil.

		The Real Part		The Imaginary Part	
Frequency (GHz)	Models	R	RMSE	R	RMSE
1.4	Dobson	0.999	0.510	0.871	2.631
	GRMDM	0.999	0.503	0.894	2.430
18.7	Dobson	0.999	1.462	0.910	2.524
	GRMDM	0.999	1.433	0.895	1.856

**Table 3 sensors-20-02806-t003:** The simulation of the imaginary part of saline-alkali soil change in response to soil moisture change.

Soil Salinity (g/kg)	Soil Moisture (cm^3^/cm^3^)	DobsonS-L	DobsonS-Ku	HQR-L	HQR-Ku	WYR-L	WYR-Ku
5	0.1	0.897	0.661	9.884	1.333	3.032	0.820
	0.2	1.684	2.161	8.964	2.686	6.112	2.493
	0.3	2.508	4.373	8.279	4.805	9.342	4.885
	0.4	3.388	7.235	8.205	7.595	12.715	7.933
	0.5	4.279	10.517	8.333	10.821	16.047	11.398
10	0.1	1.682	0.710	19.682	2.058	4.742	0.939
	0.2	3.029	2.231	17.091	3.283	10.066	2.757
	0.3	4.349	4.466	15.948	5.315	15.865	5.308
	0.4	5.686	7.298	15.393	8.025	22.070	8.525
	0.5	6.987	10.562	15.180	11.175	28.301	12.158
15	0.1	2.445	0.759	29.479	2.783	6.172	1.038
	0.2	4.334	2.299	25.489	3.883	13.528	2.987
	0.3	6.135	4.520	23.618	5.828	21.752	5.689
	0.4	7.916	7.364	22.581	8.462	30.706	9.071
	0.5	9.615	10.611	22.029	10.541	39.813	12.872
20	0.1	3.186	0.806	39.277	3.508	7.445	1.125
	0.2	5.603	2.366	33.866	4.483	16.707	3.197
	0.3	7.871	4.592	31.288	6.345	27.262	6.044
	0.4	10.084	7.431	29.770	8.904	38.906	9.588
	0.5	12.169	10.662	28.877	11.913	50.868	13.559

**Table 4 sensors-20-02806-t004:** The simulation of the imaginary part of the saline-alkali soil change in response to the soil salinity change.

Soil Moisture (cm^3^/cm^3^)	Soil Salinity (g/kg)	DobsonS-L	DobsonS-Ku	HQR-L	HQR-Ku	WYR-L	WYR-Ku
0.1	1	0.251	0.620	2.046	0.755	1.104	0.684
	5	0.897	0.661	9.884	1.333	3.032	0.820
	10	1.682	0.710	19.682	2.058	4.742	0.939
	15	2.445	0.759	29.479	2.783	6.172	1.038
	20	3.186	0.806	39.277	3.508	7.445	1.125
	25	3.893	0.851	48.879	4.219	8.591	1.203
0.2	1	0.578	2.106	1.976	2.211	2.041	2.215
	5	1.684	2.161	8.694	2.686	6.112	2.493
	10	3.029	2.231	17.091	3.283	10.066	2.757
	15	4.334	2.299	25.489	3.883	13.528	2.987
	20	5.603	2.366	33.886	4.483	16.707	3.197
	25	6.812	2.430	42.116	5.073	19.630	3.389
0.3	1	0.995	4.316	2.144	4.402	3.001	4.466
	5	2.508	4.373	8.279	4.805	9.342	4.885
	10	4.349	4.446	15.948	5.315	15.865	5.308
	15	6.135	4.520	23.618	5.828	21.752	5.689
	20	7.871	4.529	31.288	6.345	27.262	6.044
	25	9.527	4.661	38.805	6.853	32.403	6.373
0.4	1	1.498	7.186	2.456	7.258	4.007	7.374
	5	3.388	7.235	8.205	7.595	12.715	7.933
	10	5.686	7.298	15.393	8.025	22.070	8.525
	15	7.916	7.364	22.581	8.462	30.706	9.071
	20	10.084	7.431	29.770	8.904	38.906	9.588
	25	12.151	7.494	36.815	9.430	46.641	10.076

**Table 5 sensors-20-02806-t005:** The errors analysis of the dielectric constant of the saline-alkali soil.

		Real Part		Imaginary Part	
Frequency (GHz)	Models	R	RMSE	R	RMSE
1.4	Dobson-S	0.999	3.240	0.969	8.702
	HQR			−0.952	11.159
	WYR			0.972	4.508
18.7	Dobson-S	0.999	7.721	0.975	10.448
	HQR			0.976	9.086
	WYR			0.975	9.475

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
