# Peer review of "Experimental Investigation of Ground Radiation on Dielectric and Brightness Temperature of Soil Moisture and Soil Salinity"

_sensors, 2020, doi:10.3390/s20102806_

Round 1
Reviewer 1 Report
Thank you for your contribution. The presented research is very interesting, especially for the purpose of surface temperature retrieval.
In my opinion, the article is well written and all presented methods and analyses all well described. Overall, the manuscript is well organised. I have only some remarks about:
Figure 2- descriptions are illegible
Equations (4), (14), (16), (18)- editing should be corrected.
Reviewer 2 Report
The manuscript is well-conceived but it has some inadequacies. I shall highlight them coherently; therefore, the authors might improve the quality and readability of the manuscript accordingly.
- Both the abstract and the introduction sections are quite large with several repetitions. Make them succinct and compact. Please omit all the repetitions.
- I would like to see a Schematic Methodological Framework incorporating all the models detailed in the Methods section.
- Only two tables have been included in the manuscript. I would like to see another table that integrates all the remaining of the data summary which supports the graphs and results as demonstrated in the paper.
- The quality of the English language is extremely low in the manuscript, as evidenced by incorrect phrases, fewer paragraph breaks and lots of unnecessary repetitions. Please have the paper proofread from a competent authority and attach the certificate while resubmitting it.
Reviewer 3 Report
Dear authors
Manuscript has interesting results
Needs to be streamlined and easier for the reader to read and understand
English needs to be improved.
The Discussion seem not to focus on the results but goes on not very relevant tangents. Needs to be rewritten.
INTRODUCTION
Covers the key points.
L69 it should be “main” and not “mainly”
L93-93. Reword
L124-127. Not sure if needed.
You have a section Materials and methods and then Methods. This is confusing either combine or use different section titles
- Methods
You present a lot of equation. Did you use all of them? How did you combine them? Would be nice to have a diagram to show how one supplements or feeds on into the other. This would help the reader better understand the methodology.
- Results
By real part do you mean based on the actual field measurements? Tis needs to be clarified to the reader.
Figure 3. Shows significant difference from the 4 methods? Why which is the best? Would be nice to have what the actual field measurements.
Why did you use R and R2? Also, what are acceptable RMSE? Do you believe that the values you found are acceptable?
Figure 4 and 5. Some lines are all over the place? Why so different? Was this expected?
L352-358. Are these results or are these about the methods?
Figure 6. Are there R or R2.
Better connected the brightness measurements with the previous results. Do not seem very well connected.
- Discussion
L389. I believe you mean “vegetation” and NOT “vegetable”
L387-400. This is analysis sis very extensive and I am not sure that it is so relevant to the main objective of the manuscript.
The discussion does not have figures because we do not present new results. You are presenting new results and not discussing what you found!
English is poor in the discussion difficult to read and understand.
- Conclusion
The conclusions seem unrelated to the entire discussion.
Round 2
Reviewer 2 Report
Dear Authors, I am okay with the changes made.
Thank You.
Author Response
We appreciate very much for you positive comments on our manuscript.
Reviewer 3 Report
Dear authors
Thank you for addressing most of my comments.
The manuscript has substantially improved
But i wuould like to know how many soil samples you used for the calibration. Please rpovide a number
